# Superior colliculus modulates cortical coding of somatosensory information

Saba Gharaei[1,2 ✉], Suraj Honnuraiah[1,2], Ehsan Arabzadeh [1,2,3] & Greg J. Stuart[1,2,3 ✉]

The cortex modulates activity in superior colliculus via a direct projection. What is largely unknown is whether (and if so how) the superior colliculus modulates activity in the cortex. Here, we investigate this issue and show that optogenetic activation of superior colliculus changes the input–output relationship of neurons in somatosensory cortex, enhancing responses to low amplitude whisker deflections. While there is no direct pathway from superior colliculus to somatosensory cortex, we found that activation of superior colliculus drives spiking in the posterior medial (POm) nucleus of the thalamus via a powerful monosynaptic pathway. Furthermore, POm neurons receiving input from superior colliculus provide monosynaptic excitatory input to somatosensory cortex. Silencing POm abolished the capacity of superior colliculus to modulate cortical whisker responses. Our findings indicate that the superior colliculus, which plays a key role in attention, modulates sensory processing in somatosensory cortex via a powerful di-synaptic pathway through the thalamus.

[1] Eccles Institute of Neuroscience, John Curtin School of Medical Research, The Australian National University, Canberra, ACT, Australia. [2] Australian Research Council Centre of Excellence for Integrative Brain Function, The Australian National University Node, Canberra, ACT, Australia. [3] These authors jointly supervised this work: Ehsan Arabzadeh, Greg J. Stuart. ✉email: saba.gharaei@anu.edu.au; greg.stuart@anu.edu.au

The ability of an organism to attend to, and orient toward, stimuli in the environment is critical for survival. A principal neural substrate for attentional orienting movements is the midbrain structure called the superior colliculus (SC), which receives inputs from multiple sensory modalities and plays an important role in moving the eyes, head and body toward or away from biologically significant stimuli[1–3]. As evidence of its importance, the anatomical structure and input/output architecture of the SC is conserved across a range of mammalian species[4].

It is well established that SC receives direct input from the primary sensory cortices[4–9]. What is less clear is whether SC, in turn, modulates information processing in the cortex. Work in monkeys indicates that SC can modulate activity in higher order cortical areas, with visual responses in the middle temporal area (MT) of monkeys disappearing when lesions of primary visual cortex are combined with lesions of SC[10]. In contrast, visual responses in the lateral suprasylvian area in cats, which is thought to be analogous to MT in monkeys, are increased by lesions of SC[11] (although see ref. [12]). Later work showed that a functional pathway exists from SC to MT through the pulvinar in primates[13] (although see ref. [14]).

Similar to these earlier studies in primates and cats, more recent work in mice indicates that SC modulates visual responses in higher order cortical visual areas[15] as well as in the postrhinal cortex[16]. In addition, it has recently been shown that SC can also modulate responses in primary visual cortex in mice, through the dorsolateral geniculate nucleus rather than the pulvinar[17]. Together, these studies allude to the importance of the SC for visual processing in both primary and higher order visual areas in the cortex. What is not known is whether SC also modulates cortical processing of other sensory modalities.

Rodents heavily rely on their whiskers (or vibrissae) to explore and navigate the environment. Sensory information from the whiskers is processed by the whisker associated area of the somatosensory cortex, known as the primary vibrissal somatosensory cortex (vS1)[18,19]. In addition, intermediate and deeper layers of SC also receive sensory information from the whiskers, via a projection from vS1 as well as directly via the trigeminal nucleus of the brainstem[8,20,21]. While several studies have shown that SC neurons are directly activated by whisker deflections[6,22], it is not known whether activation of SC modulates coding of whisker input in vS1. This issue is the focus of the current study.

To determine if activation of SC impacts on sensory coding in vS1, ChR2 was expressed in mouse SC. We then performed extracellular and whole-cell recordings to characterize how sensory responses in vS1 were affected by optogenetic activation of SC. We find that optogenetic activation of SC modulates cortical processing of whisker responses in vS1. This effect of SC on responses in vS1 is mediated via an indirect di-synaptic pathway through POm of thalamus.

## Results

### Activation of SC modulates vS1.
To activate SC neurons optogenetically we expressed ChR2 in mouse SC (Fig. 1a). To verify that neurons were reliably driven by light, extracellular recordings were made from intermediate layers of SC neurons in vivo using multi-electrode optrodes during whisker stimulation (Fig. 1b). Receiver operating characteristic (ROC) analysis indicated that the vast majority of neurons in SC (91%; 50 out of 55; $p < 0.05$ ROC bootstrap analysis) significantly increased their action potential firing in response to brief (15 ms) optogenetic activation (Fig. 1c–e; Supplementary Fig. 1a–c). Direct activation of SC neurons by light was also verified using whole-cell recordings from SC neurons in vitro (Fig. 1f; $n = 3$). To determine if

optogenetically activated SC neurons were responsive to whisker input, we identified whisker responsive neurons located in intermediate layers of SC (1.3–2.5 mm from the surface of the brain) using whisker pad vibrations of different amplitudes (Fig. 1g; orange). Neurons in intermediate layers of SC have large whisker receptive fields and are known to respond robustly to multi-whisker movements[7,23]. ROC analysis indicated that the majority of SC neurons in intermediate layers (80%; 40 out of 50; $p < 0.05$ ROC bootstrap analysis) responded significantly to both light and whisker stimulation (Fig. 1g; green). Light activation of SC led to an upward shift of the input–output relationship of SC neurons to whisker stimuli of different amplitude (Fig. 1h; $n = 40$). Together, these experiments indicate that SC neurons processing sensory input from the whiskers can be reliably activated using optogenetics.

To investigate how activation of SC impacts on sensory coding in somatosensory cortex, whole-cell, loose-patch, and extracellular array recordings were made from vS1 while simultaneously activating intermediate/deep layers of SC optogenetically via an optic fiber (Fig. 2a). Brief optogenetic activation of intermediate/deep layers of SC (15 ms) caused increased action potential firing in vS1 neurons (Fig. 2b). Increases in action potential firing in vS1 were observed following optogenetic activation of SC with all three recording techniques (Fig. 2c; extracellular array $n = 41$; loose-patch $n = 85$; whole-cell $n = 23$; $p < 0.05$ t-test). ROC analysis of responses obtained across multiple trials indicated that ~60% of vS1 neurons (87 out of 149; $p < 0.05$ ROC bootstrap analysis) showed a statistically significant increase in action potential firing following activation of SC (Supplementary Fig. 1d, e). The median spike latency of vS1 neurons to optogenetic activation of SC was $43.9 \pm 9$ ms ($n = 72$; see "Methods" section). Increases in action potential firing during optogenetic activation of SC were seen across all cortical depths (Supplementary Fig. 1f, g).

### SC influences processing of somatosensory information in vS1.
We next determined how SC activation impacts on responses of vS1 neurons to whisker stimulation. Whole-cell, loose-patch and extracellular array recordings were made from vS1 neurons during whisker vibrations with or without optogenetic activation of SC. In these and all subsequent in vivo experiments whisker stimulation (15 ms) was presented simultaneously with optogenetic activation of SC. For each neuron, we characterized the spiking response to whisker vibrations of different amplitudes. For the vS1 neurons that responded significantly to SC activation, the median spike latency to whisker stimulation was $36 \pm 20.8$ ms (for 25 µm stimulus), $26 \pm 16.2$ ms (for 50 µm stimulus), $21 \pm 6.7$ ms (for 100 µm stimulus) and $18 \pm 7.4$ ms (for 200 µm stimulus). We then characterized the spiking response to whisker vibrations with and without SC activation (Fig. 2d). Activation of SC increased action potential firing during whisker stimulation in 80% of neurons (101 out of 127; $p < 0.05$ ROC bootstrap analysis) that were whisker responsive (127 out of 149; $p < 0.05$ ROC bootstrap analysis). Increases in action potential output following SC activation were observed across all intensities of whisker stimulation tested (Fig. 2e). On average, activation of SC caused an upward shift in the input–output relationship of vS1 neurons to whisker stimuli, with the greatest effect observed during low amplitude whisker vibrations (Fig. 2f; $n = 101$; $p < 0.05$ t-test). The effect of SC activation on whisker responses was dependent on the whisker input–output relationship, with activation of SC only enhancing whisker responses for whisker vibration amplitudes lower than that evoking the maximal response (Supplementary Fig. 2). Thus, the greatest impact of SC activation was on whisker deflections with the smallest amplitude. Together, these experiments indicate that SC activation leads to an upward shift

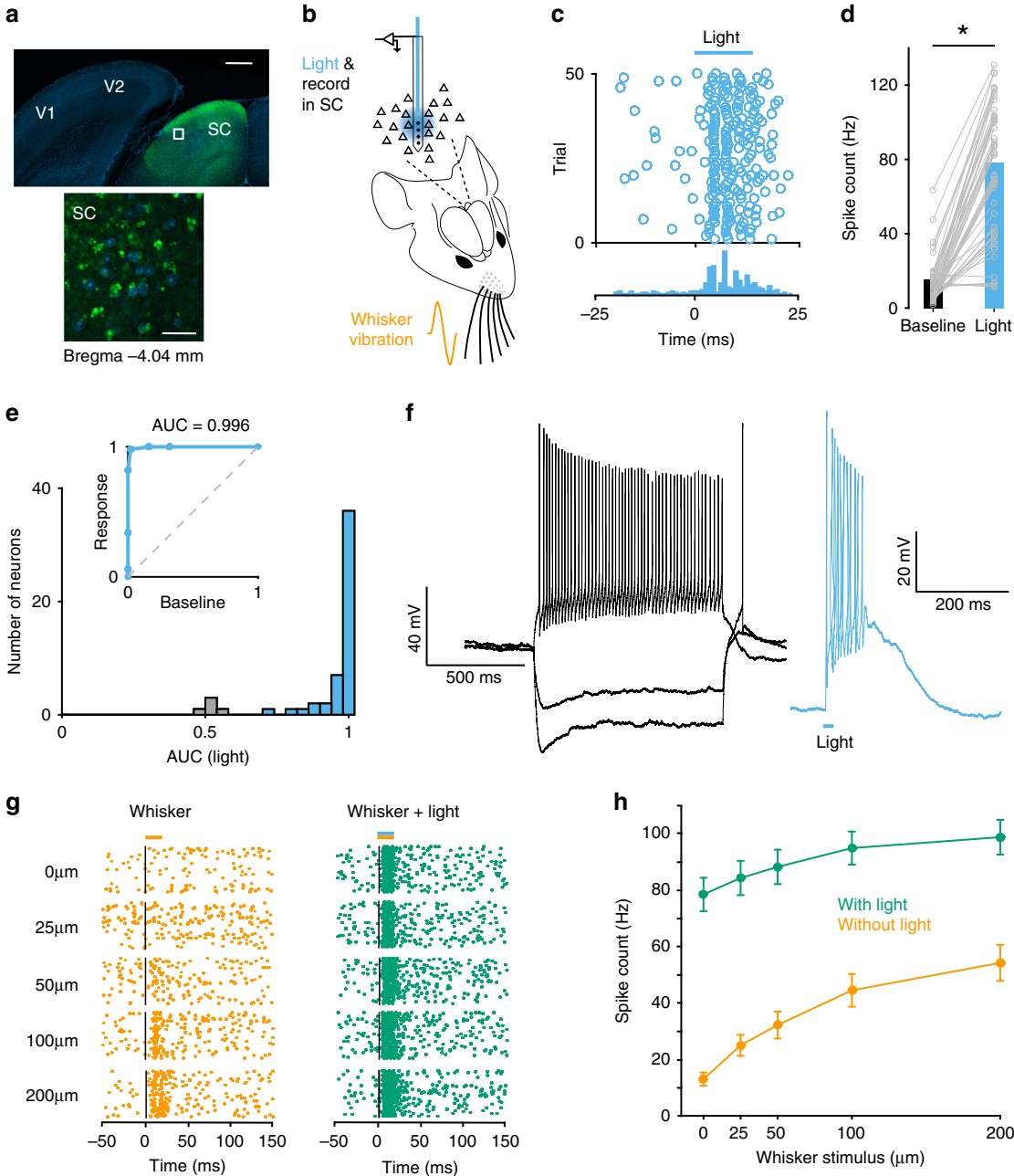

**Fig. 1 Neurons in SC are reliably activated by light and whisker stimulation. a** Coronal section showing injection site of ChR2-eYFP in SC (green). DAPI in blue. Scale bar 500 μm. SC, superior colliculus; V1, primary visual cortex; V2, secondary visual cortex. Bottom: higher magnification of the region delineated by the white square in the top image. Scale bar 25 μm. **b** Schematic of the experimental arrangement. Extracellular recording and optogenetic activation in SC with or without whisker vibration. **c** Raster plot (top) and peri-stimulus time histogram (bottom) during extracellular recording from a neuron in the intermediate layer of SC using an optrode (2289 μm from the surface of the brain) showing increased spiking in response to light activation (15 ms; blue bar). **d** Spiking activity of SC neurons ($n = 55$) increases significantly during light activation. Asterisk represents $p < 0.001$ (Two-sided paired $t$-test). **e** Inset shows the ROC curve for a representative SC neuron during light stimulation (same neuron as in **c**). The dashed line shows what is expected by chance. For this neuron, the area under the ROC curve (AUC) is 0.996. The main panel shows the distribution of AUC values for all neurons ($n = 55$). Blue bars depict SC neurons with a significant increase in spiking in response to optogenetic stimulation ($n = 50$). Gray bars depict SC neurons where there was no significant change in spiking ($n = 5$). **f** Left: voltage responses of a ChR2 expressing neuron in SC to somatic current injection (−150, −100, and 150 pA). Right: action potentials evoked in the same neuron in response to light (15 ms; LED power 0.3 mW). **g** Raster plot of action potential firing during extracellular recording from a SC neuron (2130 μm from the surface of the brain) activated by whisker movement of different amplitude alone (orange dots; left) and with light (green dots; right). **h** Pooled data showing the impact of light activation (green) on the whisker input–output relationship of whisker responsive SC neurons (orange). Only neurons that were responsive to both whisker and SC stimulation were included in this analysis ($n = 40$). Error bars represent SEM. Source data are provided as a Source Data file.

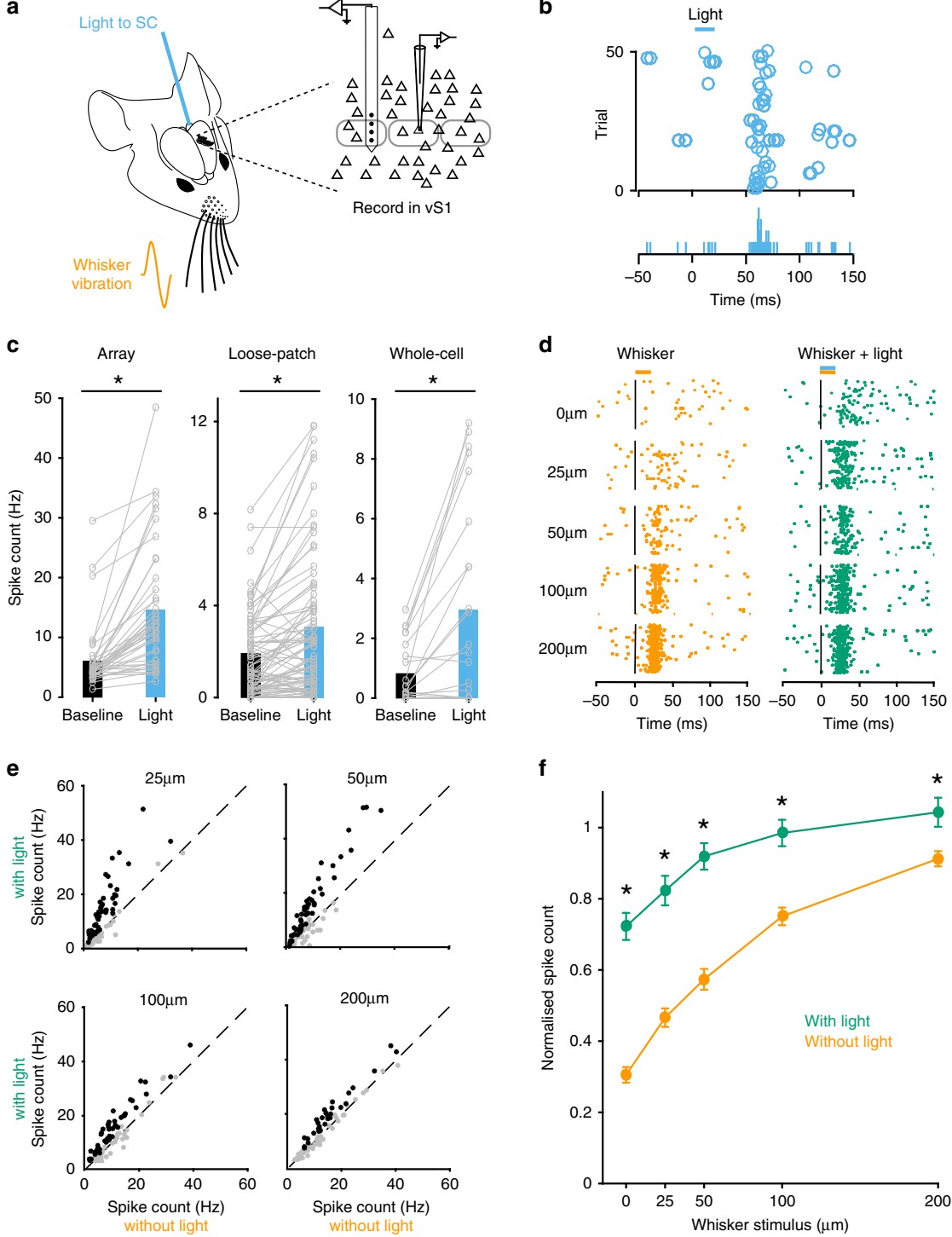

**Fig. 2 Optogenetic activation of SC modulates vS1. a** Schematic of the experimental arrangement. Recordings were made from the vS1 while activating SC optogenetically via an optic fiber in the presence or absence of whisker vibration. **b** Raster plot (top) and peri-stimulus time histogram (bottom) during loose-patch recording from a neuron in vS1 (513 μm from the surface of the brain) showing increased spiking during SC stimulation. **c** Spiking of vS1 neurons increase significantly during light activation of SC (extracellular array recordings: n = 41; loose-patch recordings: n = 85; whole-cell recordings: n = 23). Asterisks represent p < 0.001 (Two-sided paired t-test). **d** Raster plot of action potential firing during extracellular recording from a vS1 neuron (589 μm from the surface of the brain) during whisker movement of different amplitude alone (orange dots; left) and with light activation of SC (green dots; right). **e** Plot of action potential firing in whisker responsive vS1 neurons (n = 127) with and without light activation of SC during whisker stimulation of different amplitude. Black symbols indicate a significant increase in firing during SC activation (n = 101; ROC analysis). **f** Pooled data showing the impact of SC activation (green) on the whisker input–output relationship (orange). Only neurons that were responsive to both whisker and SC stimulation were included in this analysis (n = 101 neurons). Asterisks represent p < 0.001 (Two-sided paired t-test). The spiking of each neuron was normalized to the maximum response to whisker stimulation alone. Error bars represent SEM. Source data are provided as a Source Data file.

in the input–output relationship of cortical neurons during somatosensory input, resulting in enhanced responses to low intensity stimuli.

As vS1 sends a direct projection to SC[4,9], it is possible that activation of vS1 in our experiments is due to retrograde transport of ChR2 from SC to vS1. Although AAV1 is poorly retrogradely transported[24], to investigate this possibility we performed the following control experiments (Supplementary Fig. 3). Firstly, we confirmed the absence of retrogradely labeled cells in vS1 in confocal sections from animals injected with ChR2 in SC (Supplementary Fig. 3a). Secondly, whole-cell recordings from vS1 neurons in vitro in animals injected with ChR2 in SC indicated that vS1 neurons were not activated by light (Supplementary Fig. 3b, c; $n = 5$ layer 2/3; $n = 7$ layer 5). Thirdly, in animals injected with ChR2 in SC extracellular array recordings from vS1 neurons in vivo indicated that they were not activated by light when the optic fiber was moved from the SC to vS1 (Supplementary Fig. 3d, e; $n = 12$; $p < 0.05$ $t$-test). Similarly, light activation of vS1 had no impact on whisker-evoked responses in vS1 (Supplementary Fig. 3f, right). In summary, these experiments rule out the possibility that modulation of vS1 during activation of SC is due to retrograde transport of ChR2 from SC to vS1.

**Circuitry underlying the influence of SC on vS1**. We next investigated the circuitry underlying modulation of vS1 by SC. As illustrated schematically in Fig. 3a, there are two possible circuits through which SC could impact on sensory processing in vS1[2,4,8,23,25,26]. The projection from SC to the facial motor nucleus could lead to whisker movement and thereby modulate vS1 through the conventional ascending pathways via the thalamic ventral posteromedial nucleus (VPM) and posteromedial complex (POm) (Fig. 3a; pink arrows). Alternatively, an anatomical study showed projections from SC in the POm[26], suggesting that SC could modulate activity in vS1 indirectly via this pathway (Fig. 3a; purple arrows).

Previous work indicates that micro-stimulation of SC leads to whisker movement[23]. Consistent with this, optogenetic activation of SC produced whisker movements of short latency (Fig. 3b, c; average onset latency: $21.4 \pm 1.2$ ms; $n = 5$), suggesting that activation of SC could impact on sensory coding in vS1 by causing whisker movement. We, therefore, investigated whether blocking activity in the facial nerve on the same side as whisker stimulation impacted on the capacity of SC to modulate sensory coding in contralateral vS1. In rodents, motor commands driving whisker movement arise from the facial motor nucleus and project to the whiskers via the facial nerve, whereas sensory information from the whiskers is conveyed to the trigeminal nucleus via the trigeminal nerve[23,27–30]. As a result, it is possible to abolish whisker movements by blocking facial nerve activity while maintaining sensory input to vS1 from the whiskers. Whisker movements following optogenetic activation of SC were abolished by cutting or reversible cooling the facial nerve (Fig. 3d–f; $n = 3$ animals). Cutting or reversible cooling the facial nerve had no impact on baseline spiking activity of vS1 neurons (Fig. 3g, h; $n = 8$ facial nerve cut; $n = 16$ facial nerve cooling). Importantly, blocking the facial nerve also had no impact on the capacity of optogenetic activation of SC to modulate the whisker input–output relationship of neurons in vS1 (Fig. 3i; Supplementary Fig. 4). These data indicate that SC does not modulate vS1 through the generation of whisker movement. This finding is perhaps not surprising given that whisker protractions induced by SC activation are of relatively low velocity[23], and therefore not expected to have a significant impact on spiking in vS1 neurons[31].

We next investigated the possibility that SC modulates vS1 through an indirect pathway via POm. Consistent with this idea, viral injections in SC revealed that axons originating from SC target POm and not VPM of thalamus (Fig. 4a). Furthermore, we found that optogenetic activation of SC led to increased firing of POm neurons (Fig. 4b–d; $n = 38$; $p < 0.05$ $t$-test). Across the population of POm neurons recorded using both loose-patch and array recording, ROC analysis indicated that optogenetic activation of SC increased action potential firing in 66% of POm neurons (Fig. 4d; 25 out of 38; $p < 0.05$ ROC bootstrap analysis). The facial nerve on both sides of the snout were cut in these experiments, ruling out the possibility that activity in POm was driven by whisker movement. The median spike latency of POm neurons following SC activation was $23 \pm 4.0$ ms ($n = 25$). For each neuron in POm, we established a stimulus response function to whisker vibrations of different amplitude in the presence and absence of optogenetic activation of SC (Fig. 4e). SC activation significantly increased action potential firing to whisker stimulation in almost all POm whisker responsive neurons (15 out of 16; $p < 0.05$ ROC bootstrap analysis). These experiments indicate that, as seen in vS1, SC activation causes an upward shift in the input–output relationship of POm neurons to whisker input (Fig. 4f; $n = 15$). In summary, these experiments provide functional evidence that POm neurons are reliably driven by optogenetic activation of the SC.

**SC sends a direct projection to POm**. We next investigated if the SC sends a direct, monosynaptic projection to neurons in POm using whole-cell recordings from POm neurons in vitro (Fig. 5a). Brief (2 ms) optogenetic activation of SC axons evoked excitatory postsynaptic potentials (EPSPs) in 73% of POm neurons (16 out of 22 neurons). Excitatory synaptic responses were not evoked in 6 cells even at the highest LED intensity available (5 mW). The properties of POm neurons receiving input from SC were similar to those of cells that did not receive input from SC (Supplementary Fig. 5). Of the cells receiving SC input, the majority (69%; 11 out of 16) responded to SC input by generating action potentials in response to low-intensity LED light stimulation (0.8 mW), whereas the remaining cells (5 out of 16) evoked small subthreshold responses, with an average amplitude of $3.0 \pm 0.9$ mV (Fig. 5b, c). We, therefore, classified POm cells into two groups: Cells that generated suprathreshold spiking in response to low intensity LED stimulation were classified as receiving "strong" SC input, whereas cells that generated small, subthreshold EPSPs in response to low intensity LED stimulation were classified as receiving "weak" SC input. "Weak" and "strong" neurons were found within the same POm slice. The passive and active properties of POm cells receiving strong and weak input from SC were similar, suggesting they do not represent different neuronal cell types (Supplementary Fig. 6). When tested with even lower LED intensities (less than 0.4 mW) POm neurons receiving strong input from the SC generated graded changes in EPSP amplitude, but with a very different dependence on LED power compared to cells receiving weak SC input (Fig. 5d, e). We next investigated whether POm neurons receive monosynaptic input from SC. Light-evoked EPSPs remained in the presence of TTX plus 4-AP in both types of POM neurons, confirming that they receive monosynaptic input from the SC (Fig. 5f, g; strong: $n = 9$; weak: $n = 5$). To investigate whether SC also projects to VPM, we made recordings from neurons in VPM (Fig. 5h). No excitatory postsynaptic responses were observed in any VPM neurons using the highest LED intensity available (Fig. 5i, j; 5 mW; $n = 5$). Together, these data indicated that SC powerfully drives the majority of POm neurons via a direct monosynaptic projection, but does not activate neurons in VPM.

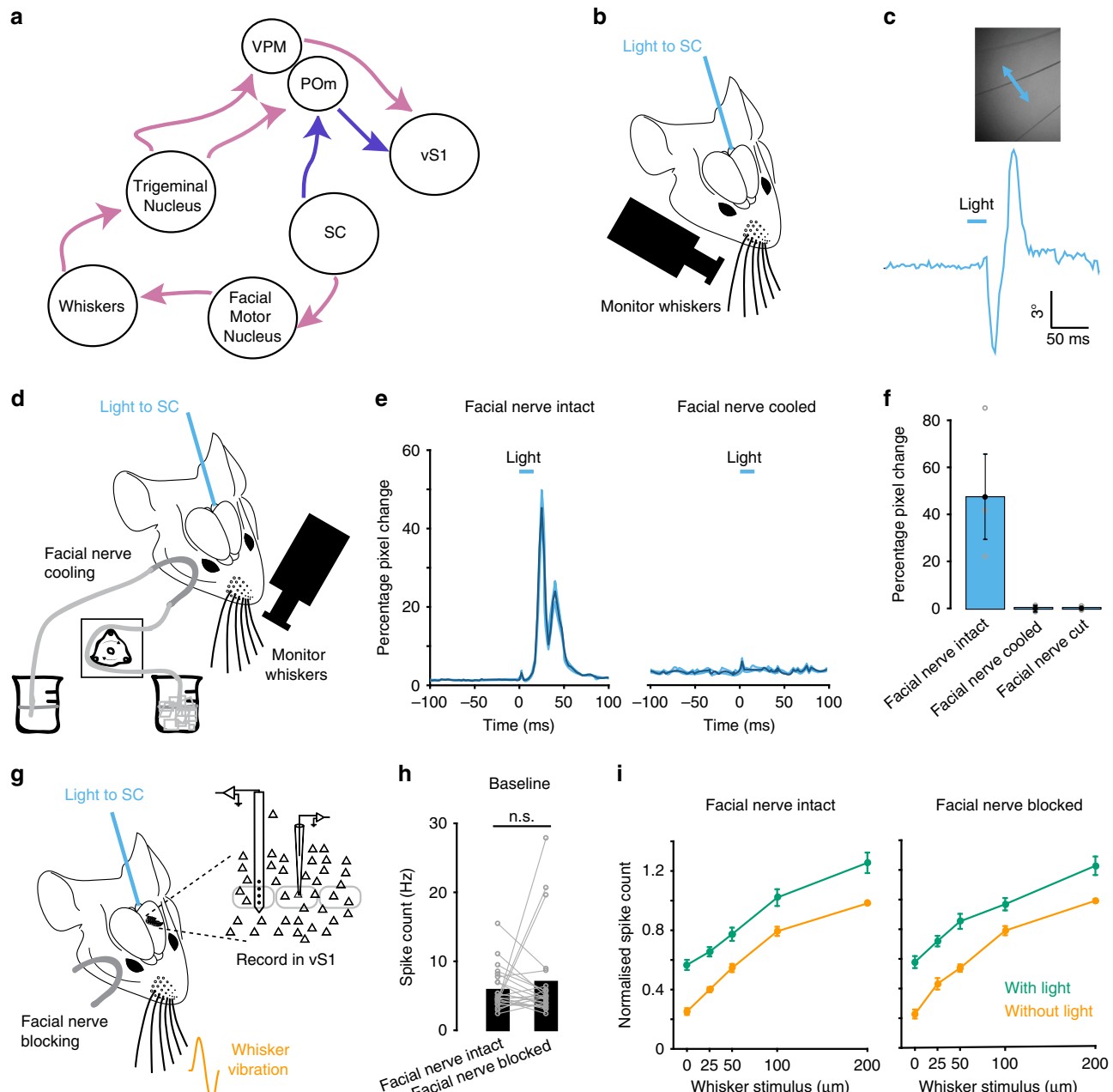

**Fig. 3 Modulation of vS1 by SC is not due to whisker movement. a** Circuits of interest showing two possible pathways through which SC could impact on responses in vS1. Modified from Castro-Alamancos and Keller[25]. **b** Schematic of the experimental arrangement using a high-speed camera to monitor whisker movement while activating SC optogenetically via an optic fiber. **c** Whisker movement during optogenetic activation of SC. **d** Schematic of the experimental arrangement. Whiskers were monitored during optogenetic activation of SC before and during facial nerve cooling. **e** Pixel change in a region of interest around whiskers during optogenetic activation of SC versus time before and after facial nerve cooling. Dark blue shows the mean and light blue the SEM across 50 trials. **f** Mean whisker pixel change ($n = 3$ animals) in response to optogenetic activation of SC before and after cutting or cooling the facial nerve. Error bars represent SEM. **g** Schematic of the experimental arrangement. The impact of optogenetic activation of SC on vS1 responses was measured before and after facial nerve block. **h** Baseline spiking of vS1 neurons does not significantly change after facial nerve inactivation ($p > 0.05$ Two-sided paired t-test; facial nerve cooling: $n = 16$; facial nerve cut: $n = 8$). **i** Pooled data showing the impact of SC activation (green) on the whisker input–output relationship (orange) for whisker responsive vS1 neurons before and after facial nerve inactivation (facial nerve cooling: $n = 16$ neurons; facial nerve cut: $n = 8$ neurons). The spiking of each neuron was normalized to the maximum response to whisker stimulation alone. Error bars represent SEM. Source data are provided as a Source Data file.

We next investigated whether SC is di-synaptically connected to vS1 through POm. To investigate this, we used an AAV-mediated anterograde trans-synaptic tagging method[5]. Cre-dependent expression of ChR2 in POm was driven by anterograde trans-synaptic expression of Cre recombinase in SC (Fig. 6a). Histological analyses revealed cell bodies expressing

eYFP fluorescence in POm (Fig. 6b, left) with light activation of these neurons leading to action potential generation ($n = 4$; data not shown). eYFP fluorescent axons were also found in vS1 (Fig. 6b, right). Whole-cell recordings from neurons in vS1 in vitro were used to determine if these POm axons provide functional input to vS1. Brief (2 ms) optogenetic activation of

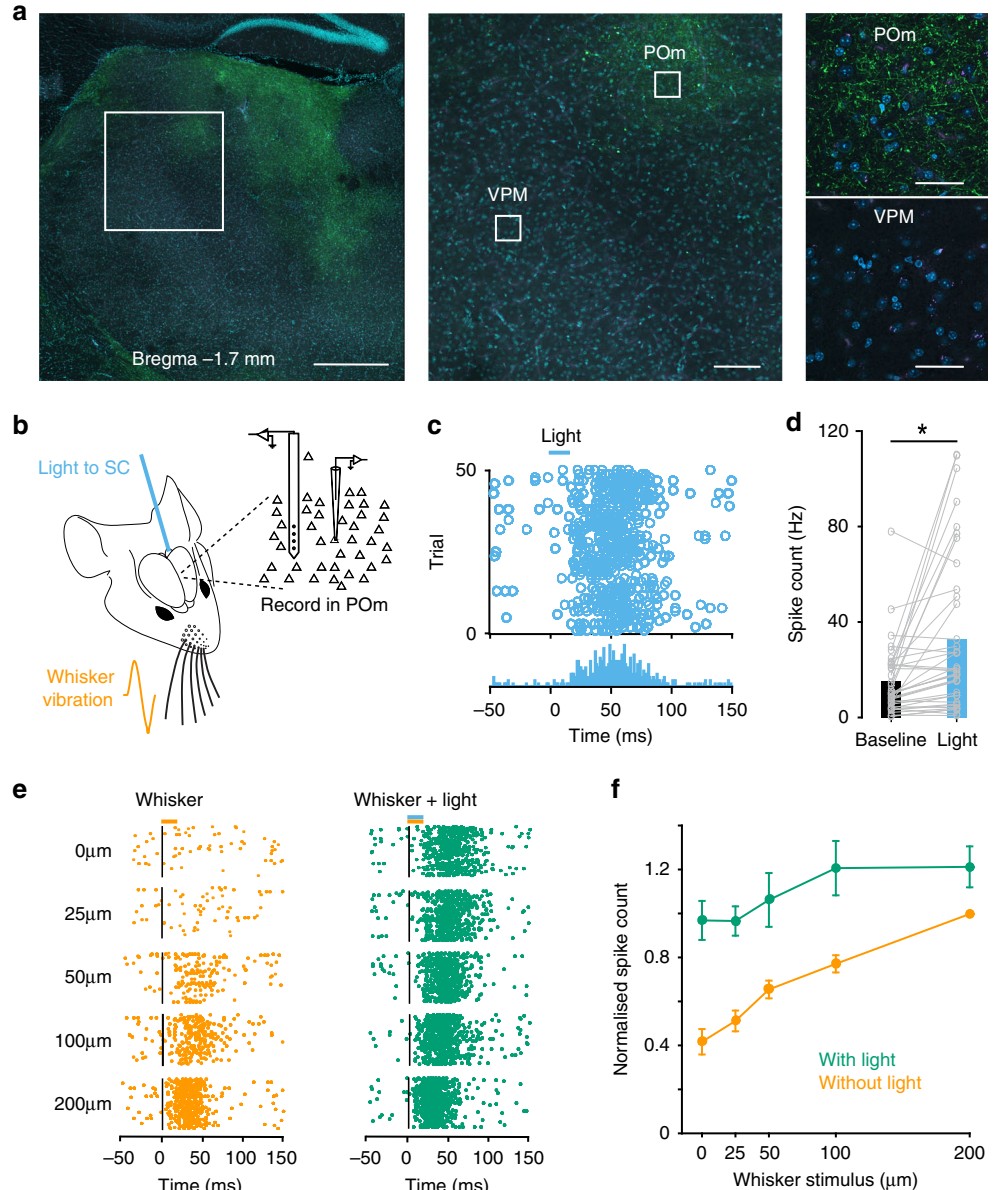

**Fig. 4 Activation of SC drives activity in POm. a** Left: coronal section of thalamus showing SC axons (in green) in POm. Scale bar 500 μm. DAPI in blue. Middle: somatosensory thalamus showing the region delineated by the white square in the left image. Scale bar 100 μm. Right: higher magnification of the regions delineated by the white squares in the middle image showing SC axons in POm but not VPM. Scale bars 40 μm. **b** Schematic of the experimental arrangement. After cutting the facial nerves, recordings were made from the POm while activating SC optogenetically through an optic fiber in the presence or absence of whisker vibration. **c** Raster plot (top) and peri-stimulus time histogram (bottom) of an extracellular array recording from a POm neuron (2407 μm from the surface of the brain) showing increased action potential firing in response to SC activation. **d** Spiking of POm neurons (*n* = 38) increases significantly during light activation of SC. Asterisk represents *p* < 0.001 (Two-sided paired *t*-test). **e** Raster plot of action potential firing in a POm neuron (same neuron as **c**) during whisker movement of different amplitude alone (orange; left) and with light activation of SC (green; right). **f** Pooled data showing the impact of SC activation (green) on the whisker input–output relationship of whisker responsive POm neurons (orange). Only neurons that were responsive to both whisker and SC stimulation were included in this analysis (*n* = 15). Spiking of each neuron was normalized to the maximum response to whisker stimulation alone. Error bars represent SEM. Source data are provided as a Source Data file.

POm axons from POm neurons receiving direct input from the SC evoked EPSPs in 80% of layer 2/3 (8 out of 10) and 100% of layer 5 neurons (6 out of 6) in vS1 (Fig. 6c–h). When tested, we found that most neurons in layer 2/3 (4 out of 5) and layer 5 (3 out of 4) receive direct, monosynaptic input from the POm as EPSPs remained in the presence of TTX plus 4-AP (Fig. 6c–h). In summary, these experiments confirm that POm neurons receiving SC input make direct, monosynaptic connections with layer 2/3 and layer 5 neurons in vS1.

**Silencing POm abolishes SC responses in vS1.** We next tested if activation of POm is required for SC modulation of vS1. To investigate this we silenced POm by pressure injection of a small volume (100–200 nl) of lidocaine into POm while recording the impact of optogenetic activation of SC on responses in vS1 (Fig. 7a). The facial nerve on both sides of the snout was cut in these experiments. Figure 7b (left) shows extracellular spiking activity of a representative vS1 neuron to optogenetic stimulation of SC. Light responses in this vS1 neuron were abolished after

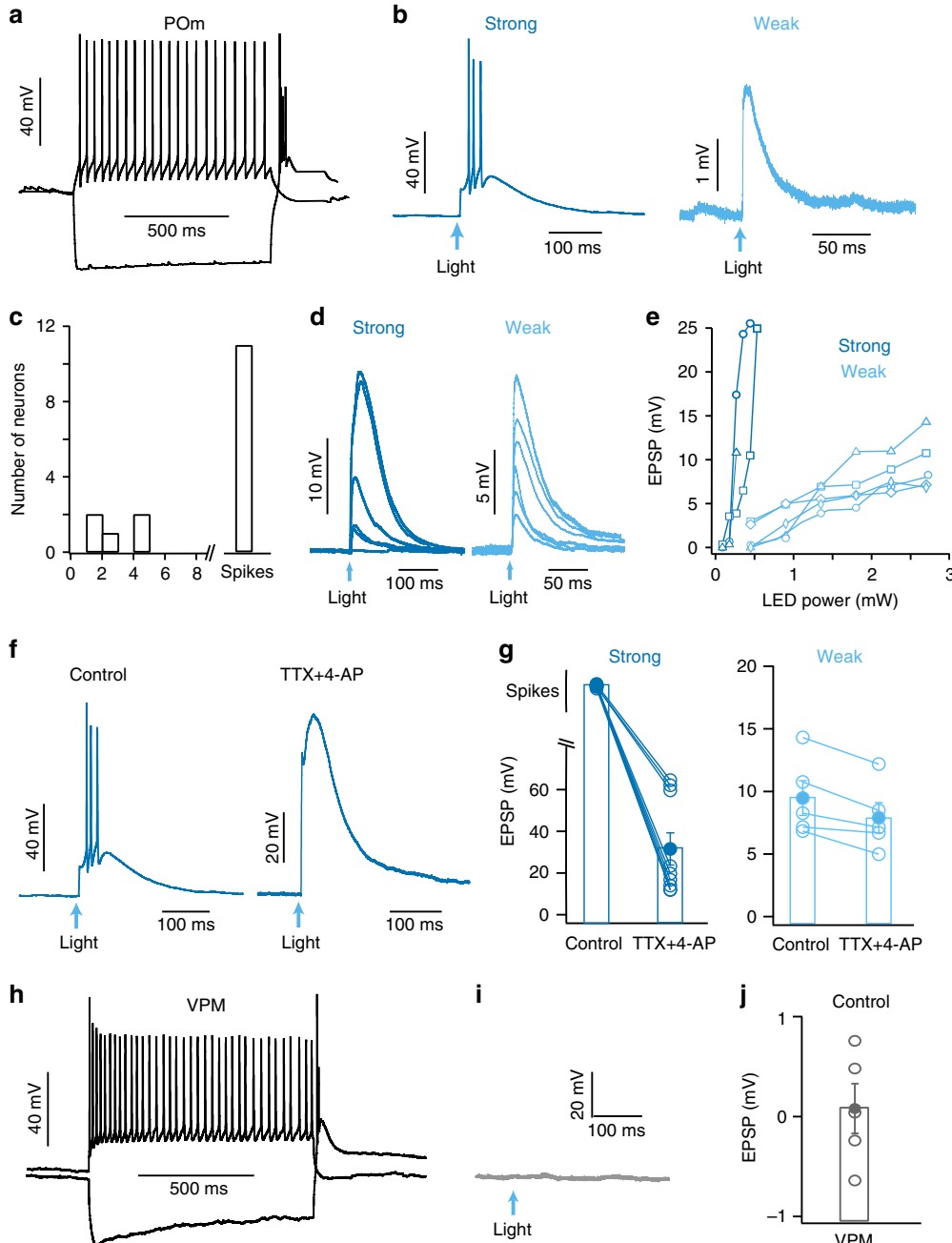

**Fig. 5 POm, but not VPM, receives direct, monosynaptic input from SC. a** Response of a POm neuron to somatic depolarizing (+300 pA) and hyperpolarizing (−400 pA) current injection. **b** Synaptic responses from POm neurons receiving "strong" (dark blue) and "weak" (light blue) SC input (LED power 0.8 mW). Arrows depict time of photo-activation of SC axons (2 ms). **c** Histogram of EPSP amplitude or presence of spiking in POm neurons during optogenetic activation of SC axons ($n = 16$; LED power 0.8 mW). **d** Graded synaptic responses in POm neurons receiving "strong" (left; LED power 0.1–0.45 mW) and "weak" (right; LED power 0.45 to 2.72 mW) input from SC. **e** EPSP amplitude in different POm neurons plotted as a function of LED power in POm neurons receiving "strong" and "weak" SC input ($n = 3$ and $n = 5$, respectively). **f** Synaptic responses in a POm neuron receiving "strong" SC input in control (left) and in the presence of TTX and 4-AP (right). **g** Summary of EPSP amplitudes or spiking in neurons that received "strong" (left; $n = 9$) and "weak" (right; $n = 5$) SC inputs in control and in the presence of TTX and 4-AP (LED power 2.72 mW). **h** Response of a VPM neuron to somatic depolarizing (+300 pA) and hyperpolarizing (−400 pA) current injection. **i** Voltage response of the same VPM neuron to photo-activation of SC axons at maximum LED power (2 ms; 5 mW). **j** Summary of EPSP amplitude in VPM neurons ($n = 5$) during optogenetic activation of SC axons at the highest LED power tested (5 mW). Error bars represent SEM. Source data are provided as a Source Data file.

lidocaine was injected into POm (Fig. 7b, right). Pooled data indicated that inactivation of POm lead to a statistically significant reduction in the response of vS1 neurons to optogenetic activation of SC (Fig. 7c; $n = 36$; $p < 0.05$ $t$-test). We next tested the impact of inactivation of POm on vS1 responses during brief deflections of the whiskers. SC activation increased action

potential firing to whisker stimulation in 79% of vS1 whisker responsive neurons in these experiments (Fig. 7d, left; 23 out of 29; ROC analysis). Silencing POm significantly reduced the impact of SC activation on these vS1 neurons during 50 μm whisker deflections (Fig. 7d, right; $n = 23$; $p < 0.05$ ANOVA interaction). Importantly, inactivation of POm had no impact on

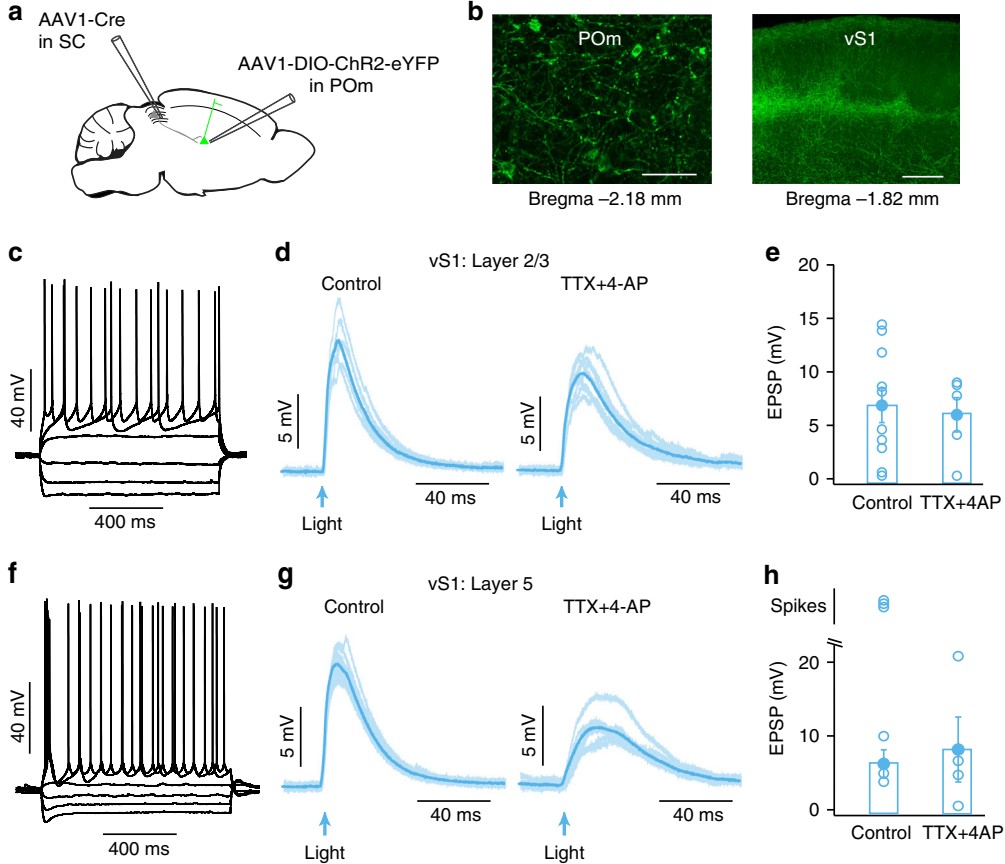

**Fig. 6 SC is di-synaptically connected to vS1 through POm. a** Schematic of the experimental paradigm for double viral injections. Expression of trans-synaptic Cre recombinase (AAV1.hSyn.Cre.WPRE.hGH) in SC was coupled with expression of Cre-dependent ChR2 (AAV1-Ef1a-DIO-hChR2(E123A)-eYFP) in POm. **b** eYFP expressing POm neurons (left: scale bar 50 μm) send axons to vS1 (right; scale bar 250 μm). **c** Response of a layer 2/3 vS1 neuron to somatic depolarizing and hyperpolarizing current injection (−400 pA to 250 pA). **d** Synaptic responses in the same neuron to optogenetic activation of POm axons (2 ms; 5 mW) in control (left) and in the presence of TTX and 4-AP (right). **e** EPSP amplitude in layer 2/3 vS1 neurons in control ($n = 10$) and in the presence of TTX and 4-AP ($n = 5$) during optogenetic activation of the axons of POm neurons receiving direct input from SC (2 ms; 5 mW). **f** Response of a layer 5 vS1 neuron to somatic depolarizing and hyperpolarizing current injection (−400 pA to 250 pA). **g** Synaptic responses in the same neuron to optogenetic activation of POm axons (2 ms; 5 mW) in control (left) and in the presence of TTX and 4-AP (right). **h** EPSP amplitude or spiking in layer 5 vS1 neurons in control ($n = 6$) and in the presence of TTX and 4-AP ($n = 4$) during optogenetic activation of the axons of POm neurons receiving direct input from SC (2 ms; 5 mW). Error bars represent SEM. Source data are provided as a Source Data file.

spontaneous baseline activity of vS1 neurons in the absence of SC activation (Fig. 7d, orange at 0 μm; $n = 23$; $p > 0.05$ t-test). Together, these results provide direct evidence that SC modulates vS1 via an indirect pathway through POm of thalamus.

## Discussion

Here we directly test the impact of SC on cortical function. While it has long been known that the cortex projects to SC, what is only now becoming clear is that SC also modulates activity in the cortex. A number of recent studies have identified the capacity of SC to influence cortical processing of visual input[16,17]. By combining optogenetic activation of SC with recordings in vS1 of mice, we show here that SC also modulates cortical processing of somatosensory input. This effect of SC on responses in vS1 was not a result of SC driving whisker movement, but instead was mediated via an indirect pathway through the POm of the thalamus.

It is well-established that SC is involved in orienting behaviors and directing attention to relevant sensory information[32], with early work by Schneider (1969) suggesting that in rodents SC is involved in the spatial localization of a stimulus[33]. The multi-sensory nature of SC together with its capacity to generate movement makes it an ideal structure for orienting responses

toward or away from a salient stimulus by initiating movement of the eyes, whiskers, head, and body. Indeed, our observation that optogenetic activation of SC in mice leads to whisker movement is consistent with earlier work showing that micro-stimulation of SC generates whisker and saccadic eye movements in rats[2,23]. Sustained whisker protractions evoked by SC stimulation are, however, different from active whisking generated by motor cortex[23]. These data suggest that the function of whisker movements caused by SC may be to position them relative to an object that has attracted the animal's attention or in anticipation of head movements rather than sensory coding[23,34,35].

Despite the importance of the SC in evoking whisker movement, our experiments show that inactivating the facial nerve, and thereby blocking whisker movement, had no impact on SC-evoked responses in vS1. Instead, we show that activation of SC drives activity in POm in vivo (Fig. 4) via a direct, monosynaptic pathway (Fig. 5). Furthermore, we show that inactivation of POm using local applications of lidocaine essentially abolishes the capacity of SC to modulate responses in vS1 (Fig. 7). These data indicate that SC modulates vS1 via an indirect pathway through POm of thalamus. This observation is consistent with an earlier anatomical study that identified projections from SC to the rostral sector of POm[26]. While it is possible that beyond blocking action

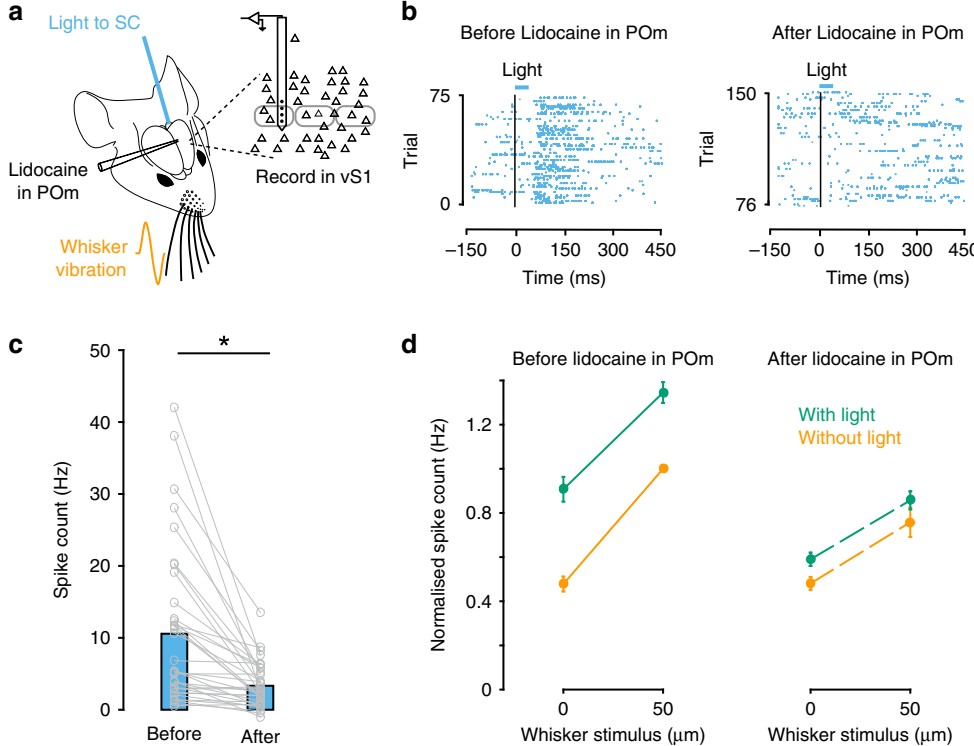

**Fig. 7 Inactivation of POm abolishes the impact of SC on vS1 neurons. a** Schematic of the experimental arrangement. After cutting the facial nerve, recordings were made from vS1 while activating SC optogenetically through an optic fiber in the presence or absence of whisker vibration. POm was silenced by local application of lidocaine. **b** Raster plot during extracellular array recording of spiking in a vS1 neuron (787 μm from the surface of the brain) in response to optogenetic activation of SC in control (left) and following inactivation of POm via lidocaine injection (right). **c** Plot of baseline-subtracted responses in a vS1 neurons to SC activation before and after lidocaine injection in POm ($n = 36$). Asterisk represents $p < 0.001$ (Two-sided paired $t$-test). **d** Pooled data showing the impact of SC activation (green) on spiking in whisker responsive vS1 neurons during small amplitude whisker movement (orange) in control (left) and following inactivation of POm following lidocaine injection (right). Only neurons that were responsive to both whisker and SC stimulation were included in this analysis ($n = 23$). The spiking activity of each neuron was normalized to the maximum response to whisker stimulation alone in control. Error bars represent SEM. Source data are provided as a Source Data file.

potentials in POm neurons, lidocaine may also suppress fibers of passage, we show that neurons in POm that receive input from the SC make direct, monosynaptic connections with layer 2/3 and layer 5 pyramidal neurons in vS1 (Fig. 6). POm also sends projections to secondary somatosensory cortex (S2)[36]. Given that S2 projects to vS1, in addition to the di-synaptic pathway described above, SC may also modulate activity in vS1 via a tri-synaptic pathway through S2.

It is well established that POm not only sends a direct input to vS1, but also modulates sensory processing in vS1[36–39]. Consistent with these studies, activation of SC increases activity of vS1 neurons, leading to larger whisker-evoked responses, particularly during small whisker movements. It is possible that during small whisker movements SC may primarily activate responses in POm, whereas during larger whisker deflections SC may recruit other structures such as zona incerta. SC is known to activate zona incerta, which has an inhibitory impact on POm[40–42]. Activation of zona incerta and subsequent inhibition of POm during large whisker movements could explain why the input–output relationship of POm neurons, and to a lesser extent vS1 neurons, plateaus during the largest whisker deflections. The idea that SC modulates activity in POm via zona incerta is consistent with recent findings showing that activation of medial prefrontal cortex modulates activity in vS1 following inhibition of POm by zona incerta[43]. At a functional level, the capacity of SC to increase whisker responses particularly during small whisker movements is expected to enhance the capacity of the cortex to direct attention to and detect weak sensory input. These data suggest

that in addition to its role in attention the SC may also play a role in feature detection.

Using whole-cell recordings in vitro, we show that SC sends a monosynaptic projection to neurons in POm, but not VPM. Interestingly, we observed three groups of cells in POm: One group received strong SC input, another weak SC input, with a third group that did not receive input from SC. These different populations of POm neurons had similar active and passive properties, suggesting they are likely to be of the same cell type. Further experiments will be required to determine the functional role of these different POm populations. Optogenetic activation of SC axons did not lead to polysynaptic responses in POm neurons that did not receive monosynaptic SC input, despite the fact that the majority of cells generated action potentials in response to SC input. This finding is consistent with earlier work indicating that POm neurons have very small or no recurrent connectivity with each other[44,45].

Thalamus routes sensory signals to the cortex and thus sits in a strategic position to modulate cortical state and the efficiency of sensory information processing[46]. Compared to VPM, POm receives weaker and more diffuse sensory projections from the trigeminal nucleus and projects mainly to layer 5a, layer 1 and inter-barrel regions of layer 4 in vS1[21,47–49]. These observations indicate that POm axons are not distributed evenly across all cortical layers, yet we found that the impact of SC activation on spiking in vS1 was not dependent on recording depth (Supplementary Fig. 1f, g). Similarly, we found that the axons of POm neurons receiving direct, monosynaptic input from SC provided

excitatory input to neurons in both layer 2/3 and layer 5 (Fig. 6). These findings are in line with earlier work showing that stimulation of POm evokes EPSPs in neurons located in all cortical layers[36].

Our observation that whisker-evoked responses in vS1 neurons are enhanced by SC through POm is consistent with earlier studies investigating the impact of POm on vS1. This earlier work indicates that whisker-evoked responses in vS1 neurons in both mice and rats are amplified by POm activation during whisker stimulation[37,38]. Sensory enhancement caused by POm is accompanied by prolongation of cortical responses over long time periods after whisker stimulation[38]. This prolonged activity in response to POm activation may prime the cortex for a behavioral response and has been shown to be critical for long-term potentiation of whisker inputs[50]. Together, with our findings, these studies suggest that the effect of SC activation on vS1 through POm may be to enhance and sustain cortical sensory signals and thereby emphasize and direct attention to salient sensory information. Consistent with this idea, enhanced population activity is observed in vS1 during simple forms of attention such as sensory prioritization[51] and temporal cueing[52]. These findings are in line with earlier work in primates[53–57] as well as attentional gain modulation seen in mouse visual cortex and thalamus[58,59].

The POm in the rodent somatosensory system is thought to be analogous to the pulvinar in the primate visual system[36]. Although the function of POm is still controversial[38,60], the role of the pulvinar in perception, selective visual attention, and visual saliency is better understood[13,61,62]. Given these findings in primates, it is perhaps not surprising that SC, which is known to be involved in attention, projects to POm in rodents. As POm not only projects to vS1, but also to other cortical areas as well as the striatum[45], our finding that POm receives direct input from SC, suggests that beyond its involvement in attention, activity in SC may generate a general priming signal that acts to modulate sensory processing in multiple brain areas.

## Methods

**Animals**. A total of 84 adult male C57BL6/J mice (age between 4 and 6 weeks) were used in this study. Mice were housed in a controlled environment with a 12-h light–dark cycle at a temperature of 22 °C and all animal procedures were approved by the Animal Experimentation Ethics Committee of the Australian National University.

**Viral injections**. Glass pipettes (Drummond), pulled on a microelectrode puller (Sutter Instrument Co.; P-87, USA) and broken to give a diameter of around 20 μm, were back-filled with mineral oil and front-loaded with viral suspension (AAV1-hSyn.ChR2(H134R)-eYFP.WPRE.hGH, AAV1.hSyn.Cre.WPRE.hGH, AAV1-Ef1a-DIO-hChR2(E123A)-EYFP; University of Pennsylvania, USA). Animals were placed in a chamber to induce light anesthesia via brief exposure to isoflurane (3.5% in oxygen) then mounted in a stereotaxic frame with anesthesia continued using isoflurane (1–1.5% in oxygen) delivered through a nose cone. Throughout surgery mice were placed on a servo-controlled heating blanket (Harvard instruments) to maintain a steady body temperature near 37 °C. For expression of ChR2 in SC, a craniotomy with a diameter of 1 mm was performed above the left SC (0.5 mm anterior to lamda and 1.5 mm lateral to the midline) and AAV1-hSyn.ChR2(H134R)-eYFP.WPRE.hGH injected into intermediate/deep layers of SC (1.5–2.0 mm from the surface; 100–180 nl; 36.8 nl per min; Nanoject II, Drummond). In experiments where ChR2 was expressed in POm neurons receiving direct input from SC, AAV1.hSyn.Cre.WPRE.hGH (200 nl) was injected into the left SC and two weeks later AAV1-Ef1a-DIO-hChR2(E123A)-EYFP was injected into the left POm (1.7 mm posterior to bregma and 1.2 mm lateral to the midline; 345–690 nl). In these double injection experiments, we did not include data from animals where there was no expression in the thalamus (the success rate of these experiments was ~50%). Following viral injection the scalp incision was closed and ketoprofen (5 mg per kg; subcutaneous) was given for pain relief. At the completion of surgery mice were returned to their cage and placed on a heating pad to recover.

**Surgery for in vivo experiments**. Three to four weeks after viral injection of AAV1-hSyn.ChR2(H134R)-eYFP.WPRE.hGH into SC, anesthesia was initially induced with brief exposure to isoflurane (3.5% in oxygen) and maintained by intraperitoneal administration of urethane (500 mg per kg) together with chlorprothixene (5 mg per kg). The level of anesthesia was regularly monitored by checking hind paw and corneal reflexes, and maintained at a stable level by administering top-up dosages (10% of original) as required. Atropine (0.3 mg per kg, 10% weight per volume in saline) was administered subcutaneously to reduce secretions. The animal was placed on a servo-controlled heating blanket (Harvard instruments) to maintain a steady body temperature near 37 °C. A custom-built head holder was glued to the skull and stabilized with dental cement. The head holder was mounted on a steel frame to minimize head movement. A craniotomy (diameter 1 mm) was performed above the left SC at the same location where viral injections had been performed (0.5 mm anterior to the lambda and 1.5 mm lateral to the midline). Another craniotomy (diameter 2 mm) was performed above the left vS1 (1.5 mm posterior to the bregma and 3 mm lateral to the midline). Saline was applied to exposed areas so they remained moist. In a subset of experiments, a craniotomy was performed above POm (diameter 2 mm, centered 1.7 mm posterior to bregma and 1.2 mm lateral to the midline). Dura mater was left intact for all areas. At the end of the experiment, the animal was euthanized with an overdose of sodium pentobarbitone (100 mg per kg intraperitoneal; Lethobarb; Verbac Australia, NSW, AUS). To confirm viral expression and electrode location, animals were perfused trans-cardially with 0.9% sodium chloride solution and then 4% paraformaldehyde (PFA). The brain was removed from the skull and kept in PFA overnight. Coronal slices (150 μm thick) were prepared, DAPI stained and examined under a confocal microscope (Zeiss LSM800 with Airyscan).

**Multi-electrode array recording**. Extracellular single unit activity was recorded with 4-channel linear Neuronexus silicon electrodes (spacing between electrodes: 100 μm). Recordings from SC were performed using an optrode (a 4-channel linear silicon electrode equipped with an optic fiber; Neuronexus) inserted vertically into SC (1.5–2.5 mm from the surface). During recordings in vS1, Neuronexus silicon electrodes were inserted at an oblique angle of 45° (0.15–1.1 mm depth). Signals from all 4 electrodes of the array were simultaneously amplified, filtered (250–5000 Hz) and were continuously recorded onto disk at a sampling rate of 40 kHz (Plexon amplifier). Data were sorted off-line to identify spiking activity on each channel. A negative threshold of 4 standard deviations of the background noise was used to detect spikes on each channel. For recordings in POm, electrodes were inserted to a depth of 2.4–3.0 mm from the surface of the brain. In a subset of these experiments, to determine the location of recording sites the multi-electrode array was dipped in fluorescent dye (Fast DiI oil; ThermoFisher Scientific) prior to insertion, with electrode location verified post-hoc.

**In vivo whole-cell and loose-patch recording**. Whole-cell[63] and loose-patch recordings were used to record the subthreshold and spiking activity of vS1 neurons. Patch pipettes were pulled from borosilicate glass and had open tip resistances of 5-7 MΩ when filled with an internal solution containing (in mM): 130 K-gluconate, 10 KCl, 10 HEPES, 4 MgATP, 0.3 Na₂GTP, 15 Na₂Phosphocreatine (pH 7.25 with KOH, osmolality ~290 mOsm). Electrodes were inserted into the brain at an oblique angle (30–45°) and lowered rapidly using a Sutter micromanipulator with high positive pressure (~200 mmHg) to pass the dura mater. The pressure was then dropped to 30 mmHg and the pipette advanced at a speed of ~2 μm per second while searching for neurons. Pipette resistance was constantly monitored in voltage clamp by applying 10 mV voltage pulses with a duration of 20 ms at a frequency of 25 Hz. For loose-patch (juxta-cellular) recordings the final seal resistance was more than 40 MΩ. For whole-cell recordings, following contact with a cell the command potential was hyperpolarized to −65 mV and constant suction of up to 70 mmHg applied. After a gigaseal was established, brief pulses of suction were applied to the pipette to rupture the membrane. Both loose-patch and whole-cell recordings were performed in current clamp using a BVC-700A amplifier (Dagan Corporation, Minneapolis, MN). Voltage was low-pass filtered at 10 kHz using a Bessel filter prior to being digitized at 40 kHz using either an ITC-18 (Instrutech) or a PCIe-6321 data acquisition board (National Instruments). Current and voltage signals were acquired by a PC computer running Axograph acquisition software (Axograph Scientific, Sydney, Australia).

**In vivo optogenetic activation**. For simultaneous recording and light activation of SC, we used an optrode equipped with an optic fiber (125 μm core diameter; Neuronexus). The optrode optic fiber was connected to a 470 nm LED (Thorlabs), with the power of blue light out of the fiber tip set to 1.8 mW. The LED was controlled through a National Instrument board using programs written in Matlab (MathWorks, Inc., Natick, MA). For activating SC while recording from vS1 or POm, photo activation was delivered through an optic fiber (200 μm core diameter) connected to a 470 nm LED (Thorlabs). In this case, the power of blue light from the tip of the optic fiber was 2.9 mW. Both the optrode and the optic fiber were inserted vertically into the SC (1.5–2.5 mm from the surface). The duration of SC light activation was 15 ms. In a subset of control experiments, optrodes (125 μm core diameter; Neuronexus) were inserted at an oblique angle of 45° into vS1 to simultaneously record and potentially stimulate vS1 neurons.

**Whisker stimulation**. Brief (15 ms) whisker deflections were applied to the right vibrissal pad (contralateral to the recording site) using a light-weight fine mesh plate glued to a piezoelectric ceramic (Morgan Matroc, Bedford, OH). The piezo was driven by an amplifier (PiezoDrive, amplification gain of 20) controlled by commands generated in MATLAB and sent to the analogue output of a PCIe-6321 data acquisition board (National Instruments; 20 kHz sampling rate). The voltage signal had a Gaussian waveform, which produced a brief deflection (6 ms rise, 9 ms decay) with minimal ringing[64]. Five amplitudes (0, 25, 50, 100, and 200 μm) were delivered in the vertical direction either with or without optogenetic activation of SC (10 stimulus conditions in total). In all in vivo experiments whisker stimulation was presented simultaneously with optogenetic activation of SC. Stimuli were applied in a pseudo-random order with 50 repetitions per condition with 850 ms inter-stimulus interval. Each episode of recording included 500 trials.

**Whisker tracking**. For whisker tracking, whiskers contralateral to the SC activation were trimmed to the level of the facial hairs except for C row, which was illuminated from below by visible light. High speed videos were captured at 400 frames per second with a high-speed camera (Mikrotron EoSens CL, Unterschleissheim Germany or a CMOS camera PhotonFocus, Lachen, Switzerland mounted on a Leica M80 stereomicroscope) during a 1 s period (0.5 s before and 0.5 s after SC activation). Frame acquisition was triggered by a National Instrument board. Each frame was then filtered using an edge detection function. Whisker movement was quantified by measuring the mean percentage change in pixel intensity from one frame to another. In some cases, we used automated software[65] to calculate whisker angle and curvature.

**Cutting or reversible inactivation of the facial nerve**. In some experiments, mice underwent bilateral facial nerve (VII) transection before performing the craniotomy and recordings. Two incisions were made in the skin covering both cheeks to expose the facial nerves. The facial nerves on both sides were then cut using microsurgical scissors under a dissecting microscope using procedures similar to that described in previous papers[28,29]. In a subset of these experiments, reversible inactivation of the facial nerve was achieved by nerve cooling. The facial nerve was exposed and a custom-made stainless steel "cryo-loop" was placed over the exposed nerve. The cooling procedure was similar to that described previously[66,67].

**Pharmacological inactivation of POm in vivo**. In some experiments, we silenced the activity of POm by pressure injection of lidocaine. A glass patch pipette (tip diameter 20 μm) was back-filled with mineral oil and then front-loaded with 10% lidocaine in ACSF. These experiments required craniotomies over SC, POm, and vS1. Extracellular recordings from vS1 neurons were performed using 4-channel linear Neuronexus silicon electrodes inserted at an oblique angle of 45° in vS1, while SC activation was achieved using an optic fiber inserted vertically into the SC. The pipette containing lidocaine was then inserted into POm. A vibrating mesh contacted most of the whiskers on the contralateral side. Control data was collected without any pressure applied to the back end of the lidocaine-containing pipette in POm. Subsequently, 100–200 nl of 10% lidocaine was injected into POm using a Nanoject to inactivate POm while recording from the same neurons in vS1.

**Brain slice recordings**. Three to four weeks after viral injection of AAV1-hSyn. ChR2(H134R)-eYFP.WPRE.hGH into SC or AAV1-Ef1a-DIO-hChR2(E123A)-EYFP into POm, mice were deeply anesthetized with isoflurane (3% in oxygen) and immediately decapitated. The brain was quickly extracted and sectioned in a chilled cutting solution containing (in mM): 110 choline chloride, 11.6 N-ascorbate, 26 NaHCO3, 7 MgCl2, 3.1 Na-pyruvate, 2.5 KCl, 1.25 NaH2PO4, 0.5 CaCl2 and 10 glucose (pH = 7.4). Coronal slices at 300 μm thickness containing either SC, POm or vS1 were prepared using a Leica Vibratome 1000S. Slices were incubated in an incubating solution containing (in mM): 92 NaCl, 2.5 KCl, 1.2 NaH2PO4, 30 NaHCO3, 20 HEPES, 3 Na-pyruvate, 2 CaCl2, 2 MgSO4 and 25 glucose at 35 °C for 30 min, followed by incubation at room temperature for at least 30 min before recording. All solutions were continuously bubbled with 95% O2/5% CO2 (Carbogen).

Whole-cell patch-clamp recordings were made under visual control from SC, POm, VPM or vS1 neurons using infrared-differential interference contrast optics[68,69]. During recording, slices were constantly perfused at ~2 ml per minute with carbogen-bubbled artificial cerebral spinal fluid (ACSF) containing (in mM): 125 NaCl, 25 NaHCO3, 3 KCl, 1.25 NaH2PO4, 2 CaCl2, 1 MgCl2 and 25 glucose maintained at 30–34 °C. Patch pipettes were pulled from borosilicate glass and had open tip resistances of 5-7 MΩ when filled with an internal solution containing (in mM): 130 K-gluconate, 10 KCl, 10 HEPES, 4 MgATP, 0.3 Na2GTP, 10 Na2phosphocreatine and 0.3% biocytin (pH 7.25 with KOH). All recordings were made in current-clamp using a BVC-700A amplifier (Dagan Instruments, USA). Data were filtered at 10 kHz and acquired at 50 kHz by a Macintosh computer running Axograph X acquisition software (Axograph Scientific, Sydney, Australia) using an ITC-18 interface (Instrutech/HEKA, Germany).

Hyperpolarizing and depolarizing current steps (−200 pA to +600 pA; intervals of 50 pA) were applied via the somatic recording pipette to characterize passive and active properties of neurons. Brain slices were continuously bathed in gabazine

(10 μM) to block inhibition mediated by GABAA receptors. Other pharmacological agents used in these experiments included tetrodotoxin (TTX; 1 μM) and 4-aminopyridine (4-AP; 100 μM), as noted in the Results. For photo-stimulation of ChR2-expressing neurons and axon terminals a 470 nm LED (ThorLabs) was mounted on the epi-fluorescent port of the microscope (Olympus BX50) allowing wide-field illumination through the microscope objective. The timing, duration and strength of LED illumination was controlled by the data acquisition software (Axograph).

**Data analysis and statistics**. Data analysis was performed using custom programs in MATLAB (Mathworks, Natick, MA) or with Axograph X. For in vitro recordings, to determine whether a neuron responded to LED stimulation, the baseline noise distribution was calculated in a 50 ms window prior to LED onset (10 trials). The light-evoked response distribution was then calculated in a 50 ms window after LED onset. A neuron was classified as receiving synaptic input if the light-evoked response distribution was statistically different from the baseline noise distribution ($p < 0.05$; $t$-test). For in vivo recordings the spiking response of each neuron was defined as the number of action potentials within a 100 ms window post stimulus onset (light in SC, whisker defection or both) averaged across 50 repetitions of each stimulus. Peri-stimulus time histograms (PSTH; 1 ms bin width) were constructed for the different stimulus conditions. Response latency was defined as the first occurrence, after stimulus onset, of two consecutive bins in the PSTH where there was a significant increase in action potential number ($p < 0.05$; $t$-test). The background, spontaneous firing rate of each neuron was calculated in a 150 ms interval before the stimulus onset. To determine if a neuron responded to a stimulus we used nonparametric, ROC analysis[70]. Formally, ROC estimates how well an ideal observer can classify whether a given spike count was recorded in one of two possible conditions: Here, the absence or presence of light /whisker stimulation (or both). In each case we compared the trial-by-trial spike count after stimulation onset with that observed prior to stimulation onset (Supplementary Fig. 1a). The overlap between these two spike count distributions was quantified by applying criterion levels ranging from the minimum to the maximum observed spike count allowing determination of the area under the ROC curve (AUC; Supplementary Fig. 1b). The statistical significance of this AUC value was determined by bootstrap analysis, with shuffled AUC values calculated 1000 times using the same experimental data randomly assigned to the experimental condition. The fraction of bootstrapped AUC values greater than the observed value indicates the $p$ value (Supplementary Fig. 1c). ROC analysis was used to determine the responsivity of vS1 neurons to whisker stimulation and/or SC activation. The impact of SC activation on the whisker input–output relationship of vS1 neurons was only quantified in neurons were there was a statistically significant increase in action potential firing during both whisker stimulation and optogenetic activation of SC.

For paired data Wilcoxon's non-parametric signed-rank test, a paired t-test or a two-way analysis of variance (ANOVA) were used to test statistical significance. Statistical significance was set at $p < 0.05$. Results are presented as average values ± the standard error of the mean (SEM), unless otherwise stated. In the figures "ns" denotes not statistically significant, whereas an asterisk denotes $p < 0.05$. For anatomical analyses of confocal images, representative images are shown based on sample sizes from 3 or more mice.

**Reporting summary**. Further information on research design is available in the Nature Research Reporting Summary linked to this article.

## Data availability
The data that support the findings of this study are available from the corresponding authors upon reasonable request. The source data underlying Figs. 1d, e, h, 2c, e, f, 3f, h, i, 4d, f, 5d, e, g, j, 6e, h, 7c, d and Supplementary Figs. 1f, 2, 3e, f, 4b, 5b–g, 6a–e are provided as a Source Data file.

## Code availability
All custom Matlab codes used for data acquisition and analysis will be made available by the authors upon request.

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

## Acknowledgements

We would like to thank Randy Bruno for encouraging us to perform the POm inactivation experiments and for technical advice. This work was supported by the Australian Research Council Centre of Excellence for Integrative Brain Function (ARC Centre Grant CE140100007) awarded to E. A. and G.J.S.

## Author contributions

S.G., E.A., and G.J.S. designed the experiments and interpreted the data. S.G. performed and analyzed the in vivo experiments. S.H. performed and analyzed the in vitro experiments. S.G. drafted the paper and all authors edited and approved the final version of the paper.

## Competing interests

The authors declare no competing interests.
