## [Peer Review File · Nature Communications]

Reviewers' comments:

Reviewer #1 (Remarks to the Author):

This is an interesting paper describing the synaptic mechanisms through which superior colliculus (SC) neurons modulate whisker-evoked responses in somatosensory cortex. The authors have used optogenetics, combined with in vitro and in vivo electrophysiology to dissect the synaptic pathway from SC via POM to S1.

The paper looks quite complete, and the conclusions are in line with the observations and are straightforward. These are interesting findings that will greatly contribute to our knowledge on the synaptic pathways for somatosensation.

I have just a few comments that do not necessarily occlude publication in my opinion.

1. Fig 2 shows that SC stimulation predominantly increases spiking in cortex. Since, the subsequent figures show that this modulation acts mainly through POM one would expect that the delta spike count is highest for the target layers of POM input, i.e. L2/3 and L5. This is not obvious from the plot.

2. It is difficult to gauge the effect of facial nerve cooling on the various whisker and spiking parameters. For example, it would be useful if the authors can provide the traces of whisker movements before and after cooling as well as after cutting the facial nerve to make sure that they efficiently abolished SC activation-induced whisker movement. For the input-output relationship graphs of Fig 3e it would be helpful if the authors show the baseline absolute spiking activity, to convince readers that in both cases (w/ and w/o light) the spiking activity level is similar.

3. I like the ROC analysis for comparing the various conditions, but think that it would be more intuitive for readers if this analysis were explained in the first main figure. The text doesn't refer to the details of the methodology until line 124 but by then it has been used for analysis of the responses of SC neurons. Furthermore, the authors indicate that the level of significance of the ROC analysis was determined using bootstraps but the results are not provided anywhere.

4. I found the quantitative support for Fig. 5b a bit poor. It only contains a single example, while this represents a major point of the paper. The authors should support the numbers that are given in the text for example with the distribution of EPSP amplitudes in TTX+4-AP.

5. Along similar lines, it would be helpful to see some form of anatomical backing of the investigated pathways. Could the authors provide histology or a visual of the Chr2-labeled projections for the various pathways, e.g. where, and how extensive are the SC projections to the POM? Is there any topographical organization of these projections inside the POM, or within the responding population of POM neurons? The same for the neurons that are receiving "weak" and "strong" inputs, are they spatially segregated?

6. The paper claims that SC-POM-S1 consists of a di-synaptic circuit. However they do not necessarily address this in full. In Fig. 5 it is shown that SC forms inputs onto POM neurons and that these synapses are boosting S1 activity, however it is not shown for the POM to S1 projections. Since POM projects in addition to other cortical areas including S2, there is a (perhaps remote) possibility that the modulation is driven by S2 neurons that feedback onto S1.

Minor:

1. In fig. 5b legend: there is an inversion of 'weak' and 'strong'

2. Fig. 3e: data from facial nerve cut and facial nerve cooling pooled, would be interesting to see them separately to see if there is a difference. This figure is confusing because on the figure it says the green is with light and the red without light, but in the legend it says the red is facial nerve cooling

and the green is facial nerve cut.

ANTHONY HOLTMAAT

Reviewer #2 (Remarks to the Author):

This study investigates the effect of SC activation on S1 activity, which would occur through a pathway from SC to POm. The paper is well written and the results are interesting. However, there is a major problem with the experimental design that undermines the findings. AAV1 was injected in the SC to express ChR2 in SC neurons. The problem is that it is well known that AAV1 spreads both anterogradely and retrogradely. The retrograde transport will cause layer 5 corticotectal cells in S1, and other cortical areas, to express ChR2. Every time the authors apply blue light in SC they are exciting both SC cells and layer 5 cortical cells that project to SC. Layer 5 corticotectal cells have extensive cortical collaterals and project to other areas, including POm. This confound explains every result in the paper. When the authors claim to be exciting SC cells, they are also exciting layer 5 cells and their many targets, including the cortex and the thalamus. Even the thalamic muscimol inactivation can be explained since it will reduce the excitability of the corticothalamic loop, and consequently cortical excitability. Although there is a pathway from SC to POm (that may change the excitability of S1), the present study does not isolate this pathway. The S1 effects observed in this study are not clearly attributable to this pathway (alone or at all).

Reviewer #3 (Remarks to the Author):

Gharaei et al. have performed experiments to demonstrate that activation of superior colliculus enhances whisker response in the mouse somatosensory cortex by optogenetic activation of superior colliculus. The authors have found that the facilitatory effect of superior colliculus is mediated by activation of the POm thalamic neurons. Overall, this is a well-performed study, but I think there are various aspects, which require further explanation/clarification:

Results:

- 1) Lines 126-127: Authors indicate, "Increases in action potential firing during optogenetic activation of SC were seen across all cortical depths (Fig. 2d)". However, most of significant responses occurred at about 400 μ m from the cortical surface. Casas-Torremocha et al. (2019; doi.org/10.1007/s00429-019-01862-4) have shown that most of POm projections occur in layer 5a. The inclusion in the figure of a Nissl representative section could facilitate the location of cortical layers.
- 2) Lines 135-137: Please, include the mean latency of whisker responses in S1 cortex to compare with the latency of superior colliculus stimulation.
- 3) Lines 146-147: It is indicated that the effect of superior colliculus stimulation depends of the amplitude of movement. Is it possible that larger movements activate other structures such as the Zona Incerta? See below in the Discussion comments.
- 4) Lines 212-213. Author's state, "No polysynaptically driven cells in POm were observed". However, in vitro is difficult to assess this conclusion because other structures that project to the POm and receive from the superior colliculus and are not included in the preparation; for example the Zona Incerta.
- 5) Lines 252-255: Why is it used lidocaine instead of muscimol? Lidocaine blocks neuronal activity and passing fibers. Thus, you cannot exclude that passing fibers from the superior colliculus to anterior thalamic nuclei may be affected. Please, include this possibility in the Discussion Section.

6) Lines 266-267: I think this phrase is incorrect "Importantly, inactivation of POm had no impact on baseline activity of vS1 neurons in the absence of SC activation (Fig. 6d, red). The response to 50 mm displacements looks smaller in presence of lidocaine in the POm. Please. Clarify this issue.

Discussion.

7) Lines 330-333: I think that Authors should mention the participation of the Zona Incerta that receive inputs from the superior colliculus and projects to POm, modulating its neuronal responses to whisker stimulation (see for example, Escudero and Nuñez 2019; doi.org/10.1016/j.neuroscience.2019.01.059). You found that response modulation dependent of the whisker movement amplitude. It is possible that smaller movements induce smaller responses in the superior colliculus than larger movements. Thus, small responses in the superior colliculus may induce responses "only" in the POm nucleus while higher responses in the superior colliculus may recruit other structures that could explain the different results according to the amplitude of the movement.

8) Lines 340-342: It is not clear to me if significant responses were observed in all cortical layers with the same frequency. Fig. 2d shows more responses about 400 mm of deep. Please, clarify this issue.

Reviewer #1 (Remarks to the Author):

This is an interesting paper describing the synaptic mechanisms through which superior colliculus (SC) neurons modulate whisker-evoked responses in somatosensory cortex. The authors have used optogenetics, combined with in vitro and in vivo electrophysiology to dissect the synaptic pathway from SC via P0m to S1. The paper looks quite complete, and the conclusions are in line with the observations and are straightforward. These are interesting findings that will greatly contribute to our knowledge on the synaptic pathways for somatosensation.

I have just a few comments that do not necessarily occlude publication in my opinion.

1. Fig 2 shows that SC stimulation predominantly increases spiking in cortex. Since, the subsequent figures show that this modulation acts mainly through P0m one would expect that the delta spike count is highest for the target layers of P0m input, i.e. L2/3 and L5. This is not obvious from the plot.

RESPONSE: We agree that if SC is modulating S1 via P0m one would expect the highest spike rates in L2/3 and L5. That the recordings originally shown in Figure 2d did not reveal this pattern may be the case for the following reasons. Firstly, estimation of laminar depth in these *in vivo* experiments is prone to inaccuracies. Secondly, we cannot be certain that we were recording from the cell bodies. Thirdly, cortical responses to SC activation may be polysynaptic. While our observation that SC evokes responses across all cortical layers is consistent with earlier work (Viaene *et al.* PNAS, 108(44), 18156–18161, 2011), as the actual depth of the cell bodies of recorded neurons is unknown in these experiments we have moved this panel to the supplementary figures (now Supplementary Fig. 1f), where we now also plot the percentage of recorded neurons that responded significantly to SC activation at different depths (Supplementary Fig. 1g). To address the issue of which cells in S1 are activated by SC we include new *in vitro* data showing, as predicted, that P0m neurons that receive input from the SC make monosynaptic connections with both L2/3 and L5 cells in S1 (Fig. 6).

2. It is difficult to gauge the effect of facial nerve cooling on the various whisker and spiking parameters. For example, it would be useful if the authors can provide the traces of whisker movements before and after cooling as well as after cutting the facial nerve to make sure that they efficiently abolished SC activation-induced whisker movement. For the input-output relationship graphs of Fig 3e it would be helpful if the authors show the baseline absolute spiking activity, to convince readers that in both cases (w/ and w/o light) the spiking activity level is similar.

RESPONSE: We now provide traces showing whisker movements before and after cooling, as well as after cutting the facial nerve (see Fig. 3e). These data are quantified in Figure 3f, which shows there are no whisker movements during cooling or after facial nerve cut. With respect to the second point made by the Reviewer, we now show that the baseline absolute spiking activity was not affected by blocking the facial nerve (Fig. 3h).

3. I like the ROC analysis for comparing the various conditions, but think that it would be more intuitive for readers if this analysis were explained in the first main figure. The text doesn't refer to the details of the methodology until line 124 but by then it has been used for analysis

of the responses of SC neurons. Furthermore, the authors indicate that the level of significance of the ROC analysis was determined using bootstraps but the results are not provided anywhere.

RESPONSE: We have now plotted the ROC analysis for the SC neurons in Figure 1 (Fig. 1e) and refer to this analysis in the text when describing this figure. We have also provided a detailed description of the steps used to calculate ROC and how we determined the significance of ROC values based on bootstrapping (Supplementary Fig. 1a-c).

4. I found the quantitative support for Fig. 5b a bit poor. It only contains a single example, while this represents a major point of the paper. The authors should support the numbers that are given in the text for example with the distribution of EPSP amplitudes in TTX+4-AP.

RESPONSE: We have now added this data to Figure 5 (see Fig. 5g).

5. Along similar lines, it would be helpful to see some form of anatomical backing of the investigated pathways. Could the authors provide histology or a visual of the ChR2-labeled projections for the various pathways, e.g. where, and how extensive are the SC projections to the POM? Is there any topographical organization of these projections inside the POM, or within the responding population of POM neurons? The same for the neurons that are receiving “weak” and “strong” inputs, are they spatially segregated?

RESPONSE: We have now added confocal images of ChR2-labeled neurons in SC (Fig. 1a), axonal projections in POM and the lack of axonal projections to VPM (Fig. 4a). We mainly found SC projections in dorsal parts of POM, with SC axonal projections found across the anterior/posterior axis. Neurons receiving weak and strong SC inputs seem to have a similar topographical organization, as we found both types of neurons within the same slice.

6. The paper claims that SC-POM-S1 consists of a di-synaptic circuit. However they do not necessarily address this in full. In Fig. 5 it is shown that SC forms inputs onto POM neurons and that these synapses are boosting S1 activity, however it is not shown for the POM to S1 projections. Since POM projects in addition to other cortical areas including S2, there is a (perhaps remote) possibility that the modulation is driven by S2 neurons that feedback onto S1.

RESPONSE: We now included data showing that POM cells that receive input from the SC make direct monosynaptic connections with cells in layer 5 and layer 2/3 of S1 (Fig. 6c-h). These data prove that the proposed di-synaptic circuit SC-POM-S1 exists. We cannot rule out that in addition a tri-synaptic pathway involving other areas, such as S2, also exists. We now discuss this possibility in the Discussion.

Minor:

1. In fig. 5b legend: there is an inversion of ‘weak’ and ‘strong’

RESPONSE: The legend for Figure 5 has been rewritten.

2. Fig. 3e: data from facial nerve cut and facial nerve cooling pooled, would be interesting to see them separately to see if there is a difference. This figure is confusing because on the figure it says the green is with light and the red without light, but in the legend it says the red

is facial nerve cooling and the green is facial nerve cut.

RESPONSE: We now plot the data for cooling and cutting the facial nerve separately (Supplementary Fig. 4). These data indicate that cooling and cutting the facial nerve had a similar impact on the input/output relationship of neurons in S1 during whisker stimulation. We apologize for the error in the figure legend. This has been corrected.

Reviewer #2 (Remarks to the Author):

This study investigates the effect of SC activation on S1 activity, which would occur through a pathway from SC to P_{Om}. The paper is well written and the results are interesting. However, there is a major problem with the experimental design that undermines the findings. AAV1 was injected in the SC to express ChR2 in SC neurons. The problem is that it is well known that AAV1 spreads both anterogradely and retrogradely. The retrograde transport will cause layer 5 corticotectal cells in S1, and other cortical areas, to express ChR2. Every time the authors apply blue light in SC they are exciting both SC cells and layer 5 cortical cells that project to SC. Layer 5 corticotectal cells have extensive cortical collaterals and project to other areas, including P_{Om}. This confound explains every result in the paper. When the authors claim to be exciting SC cells, they are also exciting layer 5 cells and their many targets, including the cortex and the thalamus. Even the thalamic muscimol inactivation can be explained since it will reduce the excitability of the corticothalamic loop, and consequently cortical excitability. Although there is a pathway from SC to P_{Om} (that may change the excitability of S1), the present study does not isolate this pathway. The S1 effects observed in this study are not clearly attributable to this pathway (alone or at all).

RESPONSE: The reviewer raises an important point. While previous work shows that AAV1 is poorly retrogradely transported (see Tervo *et al.* *Neuron*, 92(2), 372–382, 2016; their Fig. 2D), we have performed a range of control experiments to rule out that retrograde transport of ChR2 back to S1 has impacted on our conclusions, as describe below.

1. We checked anatomically whether expression in SC has retrogradely transported to the primary sensory cortices. This was not the case. In animals where ChR2 was strongly expressed in SC we did not observe expression of ChR2 in vS1 or primary visual cortex, which both project to SC (see Fig. 1a and Supplementary Fig. 3a)
2. In animals where ChR2 was expressed in SC, *in vitro* whole-cell recordings from neurons in vS1 showed that these neurons were not directly activated by light (Supplementary Fig. 3b,c).
3. In animals where ChR2 was expressed in SC, *in vivo* recordings from neurons in vS1 revealed that these neurons were no longer activated by light when the optic fiber was moved from SC to vS1 (Supplementary Fig. 3d-f). There was also no impact on whisker-evoked responses of these neurons when the light was presented in the cortex (Supplementary Fig. 3f)

We have now included a paragraph in the Results where we directly address this issue and refer the reader to the control experiments described above. These control experiments rule out the possibility that retrograde transport of ChR2 to S1 accounts for our findings.

Reviewer #3 (Remarks to the Author):

Gharaei et al. have performed experiments to demonstrate that activation of superior colliculus enhances whisker response in the mouse somatosensory cortex by optogenetic activation of superior colliculus. The authors have found that the facilitatory effect of superior colliculus is mediated by activation of the POm thalamic neurons. Overall, this is a well-performed study, but I think there are various aspects, which require further explanation/clarification:

Results:

1) Lines 126-127: Authors indicate, "Increases in action potential firing during optogenetic activation of SC were seen across all cortical depths (Fig. 2d)". However, most of significant responses occurred at about 400 μ m from the cortical surface. Casas-Torremocha et al. (2019; doi.org/10.1007/s00429-019-01862-4) have shown that most of POm projections occur in layer 5a. The inclusion in the figure of a Nissl representative section could facilitate the location of cortical layers.

RESPONSE: We now plot the percentage of recorded neurons that responded significantly to light at each recording depth (see Supplementary Fig. 1g). This figure shows that if anything neurons were less responsive to SC activation at intermediate cortical depths around 400 μ m from the cortical surface, which was unclear from the plot showing the change in spiking versus recording depth in the original manuscript. As indicated in our response to Reviewer 1, while the observation that SC evokes responses across all cortical layers is consistent with earlier work (Viaene *et al.* PNAS, 108(44), 18156–18161, 2011), as the actual depth of the cell bodies of recorded neurons is unknown in these experiments we have moved this panel to the supplementary figures (now Supplementary Fig. 1f). To address the issue of which cells in S1 are activated by SC we include new *in vitro* data showing, as predicted, that POm neurons that receive input from the SC make monosynaptic connections with both L2/3 and L5 cells (Fig. 6).

2) Lines 135-137: Please, include the mean latency of whisker responses in S1 cortex to compare with the latency of superior colliculus stimulation.

RESPONSE: We now report the spiking latency for whisker vibrations of different amplitudes.

3) Lines 146-147: It is indicated that the effect of superior colliculus stimulation depends of the amplitude of movement. Is it possible that larger movements activate other structures such as the Zona Incerta? See below in the Discussion comments.

RESPONSE: We agree it is possible that during small whisker deflections SC may primarily activate responses in POm, whereas during larger whisker deflections SC may recruit other structures such as zona incerta. Indeed, zona incerta exerts an inhibitory impact on POm and this could explain the plateau in input-output response of Pom, and to a lesser extent vS1 neurons, observed during larger whisker deflections. We thank the Reviewer for their comments and now discuss this possibility in the Discussion.

4) Lines 212-213. Author's state, "No polysynaptically driven cells in POm were observed". However, in vitro is difficult to assess this conclusion because other structures that project to the POm and receive from the superior colliculus and are not included in the preparation; for

example the Zona Incerta.

RESPONSE: We agree. We meant POM neurons were not driven polysynaptically by other neurons within POM. We cannot rule out that POM neurons might be driven polysynaptically by neurons that have their inputs cut off and are not located in the slice preparation. We now make this point clear in the revised manuscript.

5) Lines 252-255: Why is it used lidocaine instead of muscimol? Lidocaine blocks neuronal activity and passing fibers. Thus, you cannot exclude that passing fibers from the superior colliculus to anterior thalamic nuclei may be affected. Please, include this possibility in the Discussion Section.

RESPONSE: We used lidocaine as previous work had shown that it was effective in shutting down POM (see Zhang & Bruno, *Elife*, e44158, 2019). However, we agree that it is possible that beyond blocking action potentials in POM neurons, lidocaine may suppress fibers of passage to other thalamic areas. While we cannot rule out that other polysynaptic pathways could be involved in modulation of S1 by SC, we now show that a direct, monosynaptic connection exists from neurons in POM that receive input from the SC to vS1 neurons (Fig. 6c-h). We now mention the possibility that lidocaine may block fibers of passage in the Discussion.

6) Lines 266-267: I think this phrase is incorrect “Importantly, inactivation of POM had no impact on baseline activity of vS1 neurons in the absence of SC activation (Fig. 6d, red). The response to 50 mm displacements looks smaller in presence of lidocaine in the POM. Please. Clarify this issue.

RESPONSE: This sentence was referring to ‘spontaneous activity’ of vS1 neurons (that is, in the absence of whisker displacement) and not the activity during 50 μ m whisker vibrations. We have reworded this text which we hope is now clearer.

Discussion.

7) Lines 330-333: I think that Authors should mention the participation of the Zona Incerta that receive inputs from the superior colliculus and projects to POM, modulating its neuronal responses to whisker stimulation (see for example, Escudero and Nuñez 2019; doi.org/10.1016/j.neuroscience.2019.01.059). You found that response modulation dependent of the whisker movement amplitude. It is possible that smaller movements induce smaller responses in the superior colliculus than larger movements. Thus, small responses in the superior colliculus may induce responses “only” in the POM nucleus while higher responses in the superior colliculus may recruit other structures that could explain the different results according to the amplitude of the movement.

RESPONSE: We thank the Reviewer for these comments. As indicated above, we now discuss this possibility in the Discussion.

8) Lines 340-342: It is not clear to me if significant responses were observed in all cortical layers with the same frequency. Fig. 2d shows more responses about 400 mm of deep. Please, clarify this issue.

RESPONSE: We apologise for not being clear. It appears in this figure that more significant

responses occur around 400 μm presumably because there were more neurons recorded at this depth. As requested by the Reviewer, we now plot the percentage of recorded neurons that responded significantly to light at each recording depth (Supplementary Fig. 1g). As indicated above, this new figure shows that if anything, neurons were less responsive to SC activation at intermediate cortical depths around 400 μm from the cortical surface.

REVIEWERS' COMMENTS:

Reviewer #1 (Remarks to the Author):

I am happy with all of the extra data and edits to the manuscript. I think it has turned into a strong and interesting data set, with a compelling interpretation.

Reviewer #2 (Remarks to the Author):

The main concern with this study was that the AAV1 employed for ChR2 expression may lead to unwanted effects caused by retrograde transport. The authors have performed experiments that indicate that retrograde transport may not be a major concern. They looked for GFP expression in cortical cells. They also recorded from the cortex (in vivo and in vitro) while optogenetically stimulating within cortex to determine if there is ChR2 expression in cortical cells. Both of these tests came out negative, suggesting that retrograde transport may not be a major concern in the interpretation of the results. These experiments should strengthen the paper's conclusions. It would be useful if the author's clarified in the methods how light was applied in cortex. The methods indicate how light was applied in SC in vivo, but not how it was applied in cortex for the new experiments when the light fiber was moved there.

Reviewer #3 (Remarks to the Author):

All my comments have been answered positively. I have no more comments. However, there is an error in the Castejon et al., reference (page 25, line 711); The authors are only Castejon, Barros-Zulaica and Nuñez. Rudy Fishell and Wu are authors of another reference.

Reviewer #1 (Remarks to the Author):

I am happy with all of the extra data and edits to the manuscript. I think it has turned into a strong and interesting data set, with a compelling interpretation.

Reviewer #2 (Remarks to the Author):

The main concern with this study was that the AAV1 employed for ChR2 expression may lead to unwanted effects caused by retrograde transport. The authors have performed experiments that indicate that retrograde transport may not be a major concern. They looked for GFP expression in cortical cells. They also recorded from the cortex (in vivo and in vitro) while optogenetically stimulating within cortex to determine if there is ChR2 expression in cortical cells. Both of these tests came out negative, suggesting that retrograde transport may not be a major concern in the interpretation of the results. These experiments should strengthen the paper's conclusions. It would be useful if the author's clarified in the methods how light was applied in cortex. The methods indicate how light was applied in SC in vivo, but not how it was applied in cortex for the new experiments when the light fiber was moved there.

RESPONSE: We had provided this information in the Methods section "*In vivo optogenetic light activation of SC*". We have changed the title of this section to "*In vivo optogenetic activation*" to make it clearer that it includes information on optogenetic activation of both SC and vS1. In addition, we have added information on the optrode used for vS1 activation.

Reviewer #3 (Remarks to the Author):

All my comments have been answered positively. I have no more comments. However, there is an error in the Castejon et al., reference (page 25, line 711); The authors are only Castejon, Barros-Zulaica and Nuñez. Rudy Fishell and Wu are authors of another reference.

RESPONSE: We have now corrected the reference.